# Evolutionary Characteristics and Trade-Offs’ Development of Social–Ecological Production Landscapes in the Loess Plateau Region from a Resilience Point of View: A Case Study in Mizhi County, China

**DOI:** 10.3390/ijerph17041308

**Published:** 2020-02-18

**Authors:** Hang Zhang, Hai Chen, Tianwei Geng, Di Liu, Qinqin Shi

**Affiliations:** 1College of Urban and Environmental Science, Northwest University, Xi’an 710127, China; zhlove@stumail.nwu.edu.cn (H.Z.); gengtianwei1002@126.com (T.G.); lcx@stumail.nwu.edu.cn (D.L.); sqq116@stumail.nwu.edu.cn (Q.S.); 2Laboratory of Earth Surface System and Environmental Carrying Capacity, Xi’an 710127, China

**Keywords:** social–ecological production landscapes, resilience, watershed development trade-offs, the Loess Plateau region

## Abstract

Social-ecological production landscape resilience (SELPR) is a significant representation of the continuous supply capacity of landscape services. It is a quantitative assessment of the spatial-temporal evolution of SELPR under internal and external disturbances that provides a scientific basis for regional ecological environments and socio–economic development. Taking Mizhi County for the study of the Loess Plateau region, a three-dimensional (social system, ecosystem, and production system) SELPR evaluation framework was constructed. Data integration was performed using the watershed as the evaluation unit. This study quantitatively evaluated the spatial–temporal differentiation of the social–ecological production landscape (SELPs) subsystem’s resilience and the total SELPR in the study area and classified the areas from the three-system resilience combination level to achieve regional development trade-offs. The results were as follows: (1) In 2009–2018, the change in the social–ecological production landscapes pattern in Mizhi County showed a significant reduction in agricultural production landscapes, relatively stable social living landscapes, and an increase in ecological landscapes; (2) in 2009–2018, the SELPR increased by 12.38%. The spatial distribution of resilience was significantly different, showing a distribution pattern of high central and low surrounding areas; (3) the county’s watershed development zones were divided into five partitions: synergistic promotion areas, ecological restoration areas, social development areas, production optimization areas, and comprehensive remediation areas. The five types of zones have a certain agglomeration effect. In addition, the main obstacle factors affecting the SELPR of each zone are quite different. The key issues and development directions of different types of watersheds are also proposed in this paper.

## 1. Introduction

Social–ecological production landscapes (SEPLs) refer to land uses and dynamic habitat mosaics that have been shaped over the years by interactions between nature and people. They can provide services to humans and maintain biodiversity [1,2]. Currently, SEPLs are affected by multiple disturbances, such as global environmental change and urbanization development under the background of land cover/land-use change (LUCC) [3,4,5]. Land-use change in the Loess Plateau region is significant [6]. The region exhibits frequent farming activities, complex and diverse production methods, and relatively fragile ecosystems [7,8]. These conditions have greatly affected regional biodiversity conservation, community human well-being, and regional production development [9,10,11]. In a changing environment, maintaining and enhancing the social–ecological production landscape resilience (SEPLR) is the foundation of regional sustainable development [12,13]. Scientific and effective assessment of the SEPLR has become a core issue of increasing attention in contemporary geography, ecology, sociology, and interdisciplinary science [14,15,16]. Quantitative characterization and evaluation of the SEPLR evolution under internal and external disturbances are conducive to exploring the necessary maintenance of landscape services, which is of considerable significance for sustainable development [17,18].

Resilience can be understood as the capacity of systems to maintain a basic structure and function after being disturbed, which emphasizes the adjustment and adaptation of human society in the context of environmental changes and provides a basic logical framework for the study of human–land relations [19,20,21]. The definition of SELPR is the capacity of SELPs to withstand disturbances without changing the system state [12,13,22]. Compared with traditional social–ecological resilience evaluation, the outstanding characteristics of SEPLR evaluation are represented by the spatial representation of landscape system elements and the quantitative characterization of resilience heterogeneity [23,24,25]. At present, relevant research mainly focuses on two aspects. On the one hand, SELPR evaluation research mainly focuses on its time evolution. For example, Yang et al. (2015) decomposed the SELPR into three aspects (economics improvement, village community development, and landscape pattern) and selected three types of data (remote sensing image data, statistical data, and farmer survey data) to characterize resilience [26]. Wang et al. (2015) constructed indicators from the social system, economic system, and ecosystem to evaluate the tourism SELPR of Qiandao Lake in China [27]. Ciftciglu et al. (2017) took the Lefke region of Northern Cyprus as an example and adopted a participatory survey approach to construct the SELPR indicator framework from three dimensions (social systems, agricultural production systems, and ecosystems) [12]. On the other hand, relevant research is the spatial heterogeneity evaluation of SEPLR. Rescia et al. (2017) used landscape pattern indices to reflect the agricultural SEPLR at different scales [25]. Li et al. (2014) applied the ecological sensitivity index, water quality index, and vegetation coverage index to characterize the SELPR of urban wetlands [24]. Petrosillo et al. (2010) characterized the multi-scale SELPR through measurements of the multi-scale disturbance composition (amount) and spatial configuration (arrangement) [28]. It followed that the former study focuses on the characteristics of system elements, the relationship between system elements, and the evolutionary mechanism of system elements over time in resilience evaluation. However, there is less concern about the spatial differentiation of the system and its evolution. The latter research has made great progress in the use of landscape patterns and configuration indicators to quantitatively characterize the spatial heterogeneity of landscape resilience. Nevertheless, it is still insufficient in highlighting the relationships between the various subsystems of the SELPs. Therefore, based on the overall thinking of the system, focusing on the spatiotemporal differentiation of the landscape system and its evolution has become an important aspect of research on the SEPLR [10,29]. In addition, current research on the SEPLR has made breakthroughs in terms of the evaluation methods [12,30], the classification of resilience [24,25], the spatiotemporal evolution, and spatial mapping [10]. However, further consideration of the relationship between the resilience components of each subsystem to explore the regional development trade-offs is still weak in methodologies and case studies. According to the existing research, research on the classification of SEPLs mostly focuses on landscape ecological risk zoning, landscape ecological security zoning, and landscape multi-functional zoning [31,32,33]. Most of these zonings focus on macro-scales (geomorphology units, administrative units, or ecological functional units, etc.), which can well-maintain the integrity of the regional natural environment, while the landscape system elements are insufficiently addressed. Therefore, it is necessary to try to conduct management zoning research from the point of view of integrated system relationships and watershed trade-off.

A watershed is a complex natural geographical area. With the continuous development of the social-economy, natural, and man-shaped risk sources overlap in the watershed. The ecosystems at the watershed scale are subject to increasing external stress and have become one of the regions with the greatest ecological stress and risk [34,35]. The Loess Plateau region is a typical fragile area and the economic poverty contiguous area in China [36]. Under the double disturbances of the natural environment and human activities, the ecological restoration, rural revitalization, and production development of traditional farming areas in the loess hilly and gully region are equally important [37]. It is, therefore, meaningful and interesting to research the SEPLR in the Loess Plateau region. Taking Mizhi County, Shaanxi Province, China, as a case, the specific objectives of this work are as follows: (1) How to develop a suitable SEPLR assessment framework at a watershed scale and analyze the spatiotemporal evolutionary pattern of the SEPLR in Mizhi County from 2009 to 2018, and (2) according to the combined characteristics of different subsystems of SEPLs, this paper forms an SEPLs trade-off development zone and identifies the main influencing factors of each zone, to provide references for SEPLR improvement.

## 2. Materials and Methods

### 2.1. Study Area

Mizhi County (109°49′ E~110°29′ E, 37°39′ N~38°05′ N) belongs to Yulin City, Shaanxi Province, China. It is located in the middle of the Loess Plateau, on the southern edge of the Mu Us Desert, and the Wuding River passes through the north and south (Figure 1). The county has a long history and is known as the “millennial ancient county” [38]. The total catchment area of the region is 1212 km^2^. The area belongs to the typical loess hilly and gully area, with broken land, severe soil erosion, and an arid climate [39]. This county can be divided into three areas with significant differences in geomorphology: the hilly areas (ridges and hills) in the northwest, the gully areas in the east, and the flat valley areas in the middle [40]. By 2018, the county included 13 townships and 396 administrative villages, with a total population of 224,400. Natural geological disasters occur frequently in this region, especially drought and flooding. Furthermore, the main crops in the area are corn, potato, scallion, millet, and mountain fruits, which constitute a diverse production landscape [10]. Mizhi County is one of the first demonstration counties in China’s Grain for Green. Between 2009 and 2018, the areas of forest and grassland in the region increased. The ecological environment has been continuously improving. Driven by social and economic transformation, the social-economy has been developed, and infrastructure construction has been improved. Nevertheless, with the disturbances of regional natural disasters, topographical ecological constraints, and urbanization shocks, problems such as rural decline and cultivated land reclamation coexist. Under such backgrounds, taking Mizhi County as a case, this study provides a wonderful research platform for exploring the space–time evolution of the SEPLR in the Loess Plateau region.

### 2.2. Data Source and Processing

The land-use map of Mizhi County in 2009 was based on “The Second Land Use Survey” data in China (scale: 1:10,000). The land-use map of Mizhi County in 2018 was constructed from remotely sensed image data and field survey data. The first high-resolution images of Mizhi county were processed by ENVI 5.1, involving the geometric correction of imagery and imagery enhancement. The map was obtained by supervised classification and visual interpretation. According to the classification standard of land-use status (GB/T21010-2007), it was divided into seven types of cultivated land, forest land, grassland, garden land, water area, construction land, and unused land, and 25 types of secondary land use. Land-use types were divided into five categories: agricultural production spaces, social living spaces, forest land ecological spaces, pasture ecological spaces, and other ecological spaces [41] (Table 1). The Digital Elevation Model (DEM) data came from the Chinese Academy of Sciences Geographic Data Cloud. The ArcGIS 10.2 hydrological analysis module was used to extract the watershed range of the study area. A total of 253 small watersheds were divided into evaluation units. The results of the evaluation within the watershed were taken from all grid mean values [31]. The distribution map of geological disasters and the degree of vulnerability in Mizhi County was obtained from the Bureau of Land and Resources of Mizhi County. The monthly precipitation data came from the China Meteorological Science Data Service Sharing Network. Data on the grain yield, areas of three types of land (terraces, dams, and irrigated land), agricultural inputs (inputs of pesticides, inputs of fertilizers, and inputs of thin films), and other socio–economic data were derived from the “Mizhi County Social and Economic Statistical Yearbook”.

### 2.3. Methods

#### 2.3.1. SEPLR Evaluation Framework

Based on the research viewpoints of scholars such as Bergamini et al. (2013), Ciftcioglu et al. (2017), and Zhang et al. (2019), the SEPLR evaluation framework for the Loess Plateau region was constructed by considering the ecological background and socio–economic development of the study area (Table 2) [2,10,12].

The social system resilience (SR) selected 4 indicators: population density, agricultural output value, proportion of displaced population, and road density. These indicators characterized the SR evolution from 4 factors: population pressure, economic level, cultural memory, and infrastructure [12,42,43]. Population density characterized population pressure [10,27]. Building on the land-use data, the population density was obtained by using the proportion of construction land in the watershed unit as the weight and the standard value of the township population density [44]. The agricultural output value reflects the economic level [12]. Based on the proportion of the agricultural land area of watershed units, the agricultural output value of each unit was calculated by the township agricultural output value. The “cultural memory” was reflected in the “Proportion of displaced population”. The fewer migrants there were, the easier it was to retain cultural memory [45]. The level of infrastructure was characterized by road density [26,39]. Ecosystem resilience (ER) was characterized by three dimensions: landscape pattern–function–process [46,47]. Biodiversity reflects the health of ecosystems. Shannon’s Diversity Index (SHDI) was used to characterize biodiversity [2,12]. The landscape connectivity index was also considered to be critical for species survival and greatly affects landscape patterns [2,10]. SHDI and landscape connectivity indices were calculated by watershed unit in Fragstats 4.2. (FRAGSTATS-University of Massachusetts, Amherst, MA, USA), egetation coverage affected ecosystem functioning. Therefore, NDVI was chosen to represent it [24,48]. Ecological processes mainly include stress processes and protection processes in this study. The ecological stress process in the loess hilly and gully region mainly focused on non-point source pollution and soil erosion in agricultural areas [6,39,49]. According to the proportion of the cultivated land area of the watershed, combined with the total application amount of fertilizer, the fertilizer application intensity was calculated. Precipitation erosivity characterized the risk of soil erosion [50]. The precipitation erosivity was calculated according to the Wischmeier empirical formula commonly used in the general soil erosion equation (RUSLE) [51]. The “three types of land” (terraced, dam, and irrigated land) areas reflected the ecosystem protection capacity. The three types of land have unique strategic value for the ecological protection of loess hilly and gully areas [52]. The production system resilience (PR) includes three elements: the natural conditions of agricultural production, social–economic conditions, and agricultural production capacity [53,54]. Slope, elevation, and geological natural disasters were selected as stability and sporadic natural environments to characterize natural conditions of production [55,56]. The topography (slope and elevation) was an important stress factor for agricultural production in the loess hilly and gully region. The greater the slope, the smaller the potential for arable land production [50,54]. The higher the elevation, the higher the cost of agricultural production [12,57]. Production capacity was a characterization of the potential of agricultural production systems, as reflected by the grain yield and cultivated land area [15,39]. The number of labor and agricultural inputs reflected the socio–economic conditions of agricultural production [2,12].

The index layer and criterion layer weights were determined by the entropy method [58]. Specific indicator weight calculation steps were as follows:

Calculated specific data (*X*’*_ij_*) specific gravity value (*S_ij_*): (1)Sij=Xij∑i=1nXij′

Calculated index information entropy (*e_j_*): (2)$$e_{j}=\left(-\frac{1}{\ln^{n}}\right)\times \sum_{i=1}^{n}S_{ij}\ln S_{ij}$$

Calculated index difference coefficient (*g_j_*):(3)$$g_{j}=1-e_{j}$$

Determined indicator weights (*w_j_*):(4)wj=gj∑j=1mgj

#### 2.3.2. SEPLs Development Trade-Offs from the Perspective of Resilience

##### Conceptual Model for Trade-Offs’ Development

The division of trade-off development referred to the difference in resilience level and its dominant factors based on resilience assessment to better identify the advantages and disadvantages of the sub-dimension system of SEPLs. In this study, the three-dimensional magic cube method was used to classify the spatial differentiation of SEPLR [59]. Based on the results of the SEPLR evaluation in 2018, a three-dimensional magic cube and a corresponding trade-off development partition model were constructed. Among them, ER(*x*), SR(*y*), and PR(*z*) were represented by a three-dimensional space axis (Figure 2).

##### Principles and Standards of Trade-Offs’ Development

The ER, SR, and PR were classified as low, medium, and high levels, respectively. The attribute value (1~3) was set according to the distance from the node to the 3D space origin. The larger the attribute value, the higher the SEPLR. The cube was composed of a 3 × 3 × 3 three-dimensional third-order, and 27 combinations were obtained. According to the combination of the three subsystems’ resilience levels of SEPLs, this paper combined and merged them by consulting relevant expert opinions, to form a regional trade-off development partition plan (Table 3).

##### Obstacle Model

Referring to related research [60], the Obstacle Model was introduced. The order of obstacle size can determine the primary and secondary relationship of each obstacle factor and judge the degree of influence of each factor on the SELPR. The Obstacle Model calculation formula was as follows:(5)Aj=wjXij′∑i=1nwjXij′×100%
where obstacle size *A_j_* is the degree to which the *j* indicator affects the resilience, *w_j_* is the weight of the *j* indicator, and *x*′*_ij_* is the normalized value of the *j* indicator.

## 3. Results

### 3.1. Changes in the SEPLs Pattern

From the perspective of changes in various landscapes (Table 3), the area of agricultural production landscapes decreased from 657.19 km^2^ in 2009 to 625.79 km^2^ in 2018, representing a decrease of 4.78%. The space area of forest ecological landscapes, pasture ecological landscapes, and other ecological landscapes expanded rapidly, with the areas of 129.80, 336.53, and 17.24 km^2^ respectively increasing to 140.83, 338.64, and 34.15 km^2^ in 2018. The total area increased by 30.05 km^2^, with increases of 8.50%, 0.63%, and 98.09%, respectively. This is mainly due to the fact that during the research period, it was a period of “China’s Grain for Green” in Mizhi County. The government implemented a subsidy policy of the cropland retirement project and conservation program, which could guide local farmers to convert cultivated land into forests (pasture). In addition, with the stimulation of the market economy, some farmers chose to go out to work in a number of areas to maintain and improve their family’s livelihood level, so some farmers have experienced farmland and land reclamation. Due to the development of urbanization, the space for social living landscapes expanded, such as the construction of public infrastructure and service facilities such as roads, which has expanded the social living landscapes space to a certain extent.

The space area of social living landscapes increased, but the change amplitude was smaller (3.84%). The increased area was 1.46 km^2^. From the perspective of the type of social–ecological landscape transfer (Table 4), the woodland ecological landscapes were mainly transferred from the agricultural production landscapes space and pasture ecological landscapes space, and the transferred areas were 10.12 and 1.07 km^2^, respectively. The space of the pasture ecological landscape was mainly transferred from the agricultural production landscapes, and the transferred area was 5.07 km^2^. The space of other ecological landscapes was also mainly transferred from the space of agricultural production landscapes, with the largest transfer area, which was 16.87 km^2^. In addition, the space of social living landscapes transformation was small. However, agricultural production landscapes, woodland ecological landscapes, pasture ecological landscapes, and other ecological landscapes were all been transferred to social living landscapes (Table 5).

### 3.2. Spatiotemporal Evolution of SEPLR

The results of SR, ER, PR, and SEPLR were obtained based on the ArcGIS 10.2 spatial superposition method, and the results were graded by the natural breakpoint method. The spatial mapping results are shown in Figure 3.

From 2009 to 2018, the SR index changed from 0.3753 to 0.5054, representing an increase of 11.01%. The spatial distribution of the SR was presented as high in the southwest and low in the middle. In 2009, the proportion of low-grade areas was 46.45%. The medium SR area accounted for 31.98%, mainly in the southeast. The high-grade areas represented an area of 262.30 km^2^. The urbanization development of these regions has promoted the upgrading of the township industry. Additionally, the development of industries, such as rural tourism and cultural industries, increasingly improved the SR. By 2018, the low-grade area had decreased significantly, and the areas had decreased to 21.37%. The medium SR areas were basically unchanged. The high-grade areas continued to spread, and the main diffusion areas were Shigou Township and Shilipu Township. In 2009 and 2018, the ER index was 37.71% and 46.82%, representing an increase of 9.11%. In 2009, the proportion of low-grade areas was 39.60%, mainly concentrated in townships such as Shigou, Shajiadian, and Yangjiagou. The middle-grade areas accounted for 51.97%, and the distribution was relatively scattered. The high-grade areas were relatively small (11.02%) and mainly concentrated in the northwest. In 2018, the low-grade areas of Shigou Township and Shajiadian Township were significantly reduced, with a reduction area of 198.10 km^2^. The amount of fertilizer applied was small in the northwest region. Shilipu Township took the lead in carrying out the construction of silt dams and comprehensive management of small watersheds. Between 2009 and 2018, the PR index slowly increased from 0.4153 to 0.4159. The PR presents a spatial distribution with a high central and low periphery. In 2009, the high-grade areas accounted for a small amount of 9.30%. The medium PR area was 39.27%. The proportion of high-grade areas was 47.88%. The trend of PR in 2018 was not significant. Its spatial pattern also changed little. The change area of the three grades did not exceed 45 km^2^. The production conditions in the valley area are superior, which is conducive to production system development.

The SEPLR index in 2009 and 2018 was 0.3364 and 0.4602, respectively, representing an increase of 12.38%. The spatial pattern of SEPLR was obviously different, showing a distribution pattern of high central and low periphery areas. In 2009, the low-grade areas accounted for 47.88% and were distributed in the northeast and southeast. The medium-grade areas accounted for 39.27% and were mainly distributed in the northwest. The high-grade area was mainly concentrated in the middle of Mizhi County, with an area of 151.46 km^2^. In 2018, the SEPLR maintained the spatial pattern in 2009, but the grade areas changed a lot. The low-grade areas were reduced by 211 km^2^. The reduction areas were located in the northeast of Shajiadian Township and Yindou Township. The areas of a medium grade increased by 159.26 km^2^. The areas of high SEPLR increased little, mainly in Guoxingzhuang Township.

### 3.3. Watershed Development Trade-Off

To better coordinate the regional ecological environment, social economy, and agricultural production, it is necessary to balance the development relationship between the ecological system, social system, and production system in the SEPLs. Taking the watershed as the basic evaluation scale, from the level of resilience of the three subsystems of SELPs, the development trade-offs of Mizhi County can be divided into the following five categories: synergistic promotion areas, ecological restoration areas, social development areas, production optimization areas, and comprehensive remediation areas. Synergistic promotion areas included 94 watersheds, the largest proportion of total regions (27.26%), which were mainly distributed among Shilipu Township, Guoxingzhuang Township, and the townships in the northeast. Ecological restoration areas included 26 watersheds, the proportion of the area of which was 12.34%, which were mainly distributed among the junction of Shigou Township and Yinzhou Township. Social development areas included 31 watersheds, the proportion of the area of which was 11.74%, which were generally distributed in the northwest containing Longzhen Township and Qiaohecha Township. Production optimization areas included 54 watersheds, the proportion of the area of which was 28.58%, which were mainly distributed in the marginal areas of Mizhi County. Comprehensive remediation areas included 48 watersheds, the proportion of the area of which was 20.05%, which were mainly distributed in Gaoqu Township, Yangjiagou Township, and Jijiacha Township. The spatial layout of the five types of partitions had a certain agglomeration effect (Figure 4).

Studying the factors affecting the SELPR and its changes in degree, nature, and attributes has important guiding significance for improving resilience and achieving regional sustainable development. According to Formula (2), the top five significant obstacle factors were calculated and screened by the degree of obstacles. The results show that there were differences in the factors that hinder the SELPR in 2009–2018 (Table 6).

Synergistic promotion areas have an excellent ecological environment, a relatively sound agricultural production infrastructure, a relatively complete industrial structure, and a high social and economic level of farmers. The obstacle factors affecting the SEPLR in the region were the amount of labor (***Z***_16_), fertilizer application intensity (***Z***_8_), slope (***Z***_11_), agricultural output value (***Z***_2_), and population density (***Z***_1_). The sum of the contributions of the first five factors was 31.21%. It can be seen that the basic conditions and production potential of agricultural production are important factors influencing the SEPLR in the region. Therefore, these watersheds should be based on the development of agricultural production, while also focusing on maintaining coordination of agricultural production, the social economy, and the ecological environment.

In the ecological restoration areas, the top five obstacle factors were the fertilizer application intensity (***Z***_8_), “three types of land” areas (***Z***_10_), precipitation erosivity (***Z***_9_), road density (***Z***_4_), and geological disaster risk (***Z***_13_) (Table 6). The sum of the contribution of the five obstacle factors was 43.78%. The region has certain advantages in terms of the high level of social–economic development, good basic conditions for agricultural production, and abundant cultivated land resources. However, the excessive use of agricultural fertilizers in large-scale farming areas in the region has led to serious environmental pollution and the significant fragmentation of landscapes, which is not conducive to the improvement in ecosystem resilience. Therefore, these watersheds should focus on ecological environment construction, pay attention to the implementation of the Grain for Green policy, reduce the farmland, reduce the agricultural fertilizer pollution, monitor the river flood season, build silt dams, and repair sick dams, to reduce the frequency of floods.

In the social development areas, the top five obstacle factors were the agricultural output value (***Z***_2_), proportion of displaced population (***Z***_3_), NDVI (***Z***_7_), amount of labor (***Z***_16_), and grain yield (***Z***_14_) (Table 6). The agricultural output value (***Z***_2_) and proportion of displaced population (***Z***_3_) were the main factors limiting the development of SEPLR in the region. The contribution of the two factors was 10.07% and 8.93%, respectively. The ecological environment in the region was well-maintained, and was implemented earlier in the construction of dams and terraces. The regional “three types of land” areas accounted for a relatively high proportion. The agricultural production was developing well and was a typical grain production base in the county. However, the development of the regional socio–economic system has yet to be further improved. The main problems were that the outflow of the local population which has led to the hollowing out of the countryside and the aging phenomenon. In addition, the region has not made good use of excellent agricultural production conditions to achieve the transformation and upgrading of traditional rural industries. In the process of regional trade-off development, the region should attach great importance to industrial restructuring; promote industrial-scale and efficiency development, especially the development of rural tourism and cultural industries, and maintain the inheritance of regional rural culture.

In the production optimization areas, the top five obstacle factors were the cultivated land area (***Z***_15_), amount of labor (***Z***_16_), slope (***Z***_11_), road density (***Z***_4_), and precipitation erosivity (***Z***_9_) (Table 6). The top five factors are all related to agricultural production. The implementation of the Grain for Green policy and the improvement of soil erosion have been outstanding in the region. The regional ecological environment is beautiful. However, there are a few problems in the region, such as remote areas, severely abandoned land, lagging agricultural infrastructure construction, irregular land plots, and poor traffic accessibility. The sum of the obstacles of the five factors was 36.84%. The district should focus on improving the utilization of cultivated land resources and agricultural industrialization, paying attention to agricultural production technology innovation and the improvement in agricultural arable land efficiency, improving road infrastructure, strengthening farmland protection measures, and promoting the development of large-scale and modern agriculture.

In the comprehensive remediation areas, the top five obstacle factors were the NDVI (***Z***_7_), road density (***Z***_4_), agricultural output value (***Z***_2_), amount of labor (***Z***_16_), and “three types of land” area (***Z***_10_) (Table 6). The total contribution of the five obstacle factors was 29.07%. Due to the different geographical environments, natural endowments, and socio–economic development levels, the region displays a considerable difference in the combination of subsystems of SEPLs. Therefore, the region should strengthen the comprehensive integration of natural environmental factors, socio–economic factors, and agricultural production factors. At the same time, it should pay attention to regional characteristics and factor advantages and promote the optimal allocation of regional resource elements.

## 4. Discussion

Resilience assessment, as an instrument that can efficaciously support the management of SEPLs, has become a vital research subject of geography and landscape ecology [12,13,14]. It has been recognized by many scholars to construct a multi-level indicator framework to characterize the social–ecological system resilience. However, the paradigm of the resilience evaluation index system has not yet been formed [10,12,26,27]. This paper drew on the research framework of SEPLR proposed by Ciftcioglu (2017). Then building on the previous framework of social–ecological system resilience, this paper constructed an evaluation framework of SEPLR from three dimensions (social system, ecosystem, and production system). Because the study area was a typical traditional agricultural planting county, agricultural production activities were the main form of human activities that shaped the local landscape. Changes in agricultural production activities could be reflected through the change in land use. The change in land use was a natural bridge that connects micro-human activities with the succession of macro-landscape ecosystems and effectively integrates them closely [58]. Therefore, assessing the SEPLR from three levels (ecosystem, social system, and production system) has some scientific and practical basis. In addition, this paper focused on combining social–economic data and agricultural production data with land-use data to produce watershed-scale data. This afforded a method for the enrichment of information in the regional SEPLR quantification process. Therefore, the index selection not only considered the system attributes of SEPLs but also paid attention to the spatial quantization of the index of each subsystem dimension. In addition, the construction of the indicator system in this paper was unique and regional. For example, the area of “three types of land” was used to characterize the ecological protection of the watershed, the intensity of chemical fertilizer application was used to characterize the ecological stress of the watershed, and the elevation and slope were used to characterize the natural conditions of agricultural production. Therefore, the evaluation indicators of SEPLR in this paper were only used in similar loess hilly and gully areas.

At present, it was common to divide the resilience unit cell into a grid-based on equally spaced sampling [4,10,24]. In addition, there were also divisions based on watersheds or administrative areas [26,31,33]. Among them, the grid-based division was conducive to the expression of the spatial heterogeneity of resilience, and the grid size could be adjusted flexibly. However, the existence of scale effects increased the uncertainty of the results. Division of units based on administrative districts can facilitate policymakers in formulating adaptive management policies and promote optimal resource allocation. However, it was easy to split the integrity of the landscape distribution and it was difficult to characterize the spatial differences in the resilience within the administrative area. As an elementary assessment unit for multi-data source integration, watersheds can ensure the integrity of the landscape system structure, function, and process, while ensuring the natural heterogeneity of diverse evaluation units and avoiding the fragmentation of natural element correlations. What is more, the differences in internal factor conditions between different watersheds were relatively obvious [31,33]. The application of the statistical unit of a small watershed is consistent with the natural environment of the loess hilly and gully area [40]. The division of such evaluation units is helpful for the comprehensive analysis of the spatial pattern of SEPLR. This method of resilience unit division can effectively analyze the resilience type and accurately grasp the management divisions for maintaining and improving resilience.

According to the current research on the SEPLR, the study paid more attention to the evaluation of SEPLR, the spatiotemporal evolution of SEPLR, the classification of resilience types, and the prediction of resilience [10,16,24]. However, there is little research on the specific suggestions for regional trade-off development management based on the resilience of different subsystems of SEPLs. Therefore, it is often more difficult for the results of resilience assessment to provide more specific applications. According to the resilience of the three subsystems, the management division of different watersheds was carried out. Then, the dominant factors of SEPLR evolution in different watersheds were identified. Finally, differentiating resilience management measures and countermeasures were proposed. This study reflected the practical application of the evaluation results of SEPLR. In some ecologically fragile areas in the world, there are trade-offs between the development of regional modern agricultural production, the fragile ecological environment, and regional socio–economic development, making regional sustainable development extremely difficult [61]. Due to the unique geomorphological environment and surface water and frequent human activities, as well as the diverse agricultural production methods in the Loess Plateau, the regional SEPLs are facing multiple disturbances [6]. Therefore, it is very important to study the evolution and development trade-off of the resilience of the landscape system in this area.

However, in this study, there were still some limitations. In particular, it characterized the accuracy and subtlety of multidimensional indicators of resilience. In further research, it will be necessary to collect comprehensive and accurate data to quantify the characterization more effectively under the premise of coordinating a multi-factor and multi-dimensional assessment of SEPLR. Moreover, to effectively improve the SEPLR at a watershed scale, it is necessary to clarify its internal constraints, and integrally analyze the comprehensive impact mechanism of SEPLR from multiple perspectives. Finally, to propose a more systematic development strategy for regional trade-offs, a more scientific and standardized quantification method was formed. The discussion of the spatiotemporal evolution trend of the SEPLR at the macro level and its coordinated management zones only provided analysis of the macro development trend of the SEPLR in the study area and the regional adaptive management method at the watershed level, while accurately providing countermeasures and measures to improve the SEPLR required further revealing the driving mechanism of the SEPLR evolution from a micro-level. This provided the foundation for the multi-scale study of the SEPLR in the future. This paper has not systematically explored the evolution and driving mechanism of SEPLR at a micro-scale. Consequently, the focus of future in-depth research will be to explore the spatiotemporal evolution of SEPLR at multiple scales, the coupling relationship between the SEPLR and the subsystem resilience and pay attention to the impact of changes in land-use behavior of micro-subjects (farmers) on resilience.

## 5. Conclusions

This study constructed a three-dimensional resilience assessment framework, which can effectively reflect the spatiotemporal evolution of SEPLs. From the perspective of a combination of the resilience of the three subsystems of SEPLs, the types of regions were divided to achieve regional development trade-offs. The research results were as follows:From the perspective of changes in landscapes, the landscape area of agricultural production decreased by 4.78%. The forest ecological landscapes, pasture ecological landscapes, and other ecological landscapes areas expanded rapidly, and the total area increased by 30.05 km^2^. The area of social living landscapes also increased, but the extent of change was small. From the perspective of transfer type, the forest ecological landscapes was mainly transferred from agricultural production landscapes and pasture ecological landscapes, and the space of pasture ecological landscapes was mainly transferred from agricultural production landscapes;During the period of 2009–2018, the ER index steadily increased from 0.3771 to 0.4682. The ER presents a spatial pattern that was high in the northwest and low in the southeast. The index of SR rapidly rose from 0.3753 to 0.5054. The SR presents a spatial pattern that was high in the southwest and low in the middle. The PR index slowly increased from 0.4153 to 0.4159. The spatial pattern of PR appeared to be high in the middle and low in the periphery. During the study period, the spatial differentiation of SEPLR was significant. It presents a spatial pattern in which the central river valley was a high-value area and the periphery was a low-value area. The SEPLR steadily increased by 12.38%;The watershed SEPLs trade-off development division was divided into five categories: the synergistic promotion areas, ecological restoration areas, social development areas, production optimization areas, and comprehensive remediation areas. The obstacle factors of each partition exhibited certain differences. Based on this, this paper proposed different trade-off development suggestions for different watershed management zones.

## Figures and Tables

**Figure 1 ijerph-17-01308-f001:**
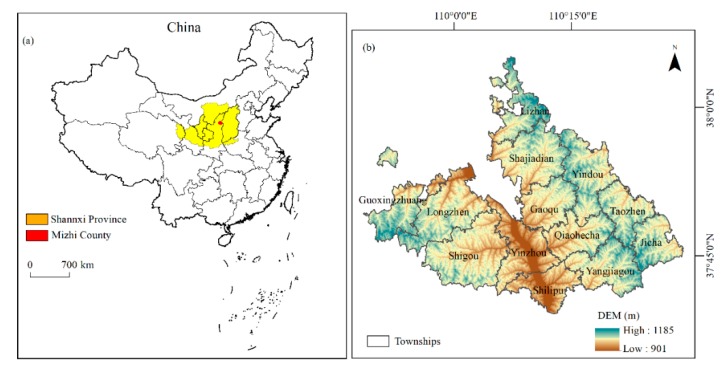
Location and DEM (Digital Elevation Model) of Mizhi County, Shaanxi Province, China.

**Figure 2 ijerph-17-01308-f002:**
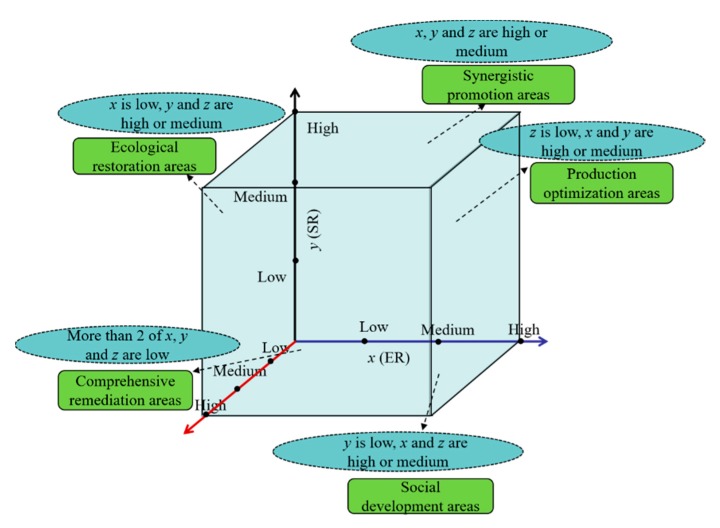
Magic cube of the connotation property of social–ecological production landscape resilience (SEPLR) and partition conceptual model.

**Figure 3 ijerph-17-01308-f003:**
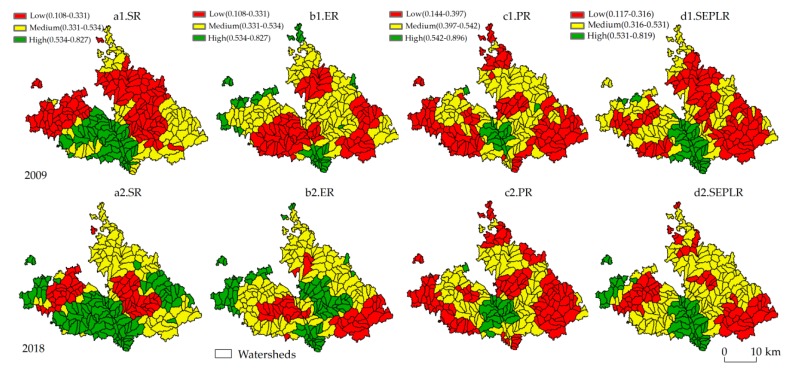
Spatial distribution of SEPLR from 2009 to 2018 in Mizhi County.

**Figure 4 ijerph-17-01308-f004:**
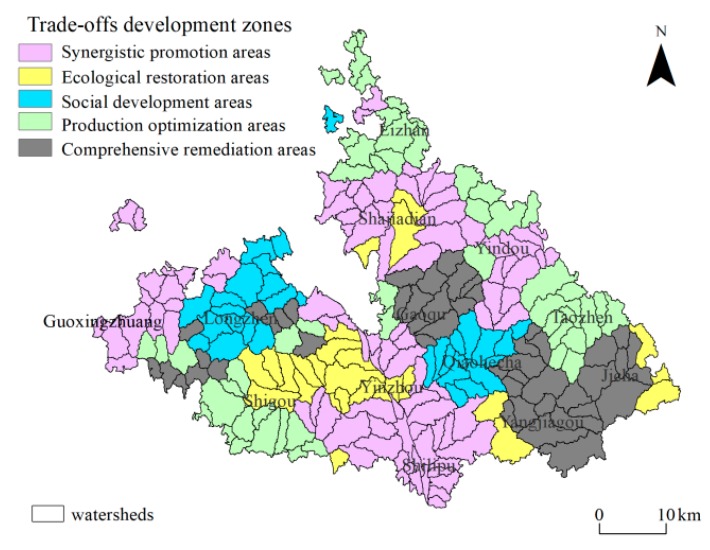
Integrated trade-off zoning at a watershed scale.

**Table 1 ijerph-17-01308-t001:** Land-use function classification.

Dominant Function Classification		Secondary Classification
First Class Ground Category	Secondary Class Ground Category	
Production land	Agricultural production landscapes	Paddy field, Dry land, Irrigated land
Living land	Social living landscapes	Constructed towns, Mining land, Scenic spots and special land, Hydraulic construction land, Railway land, Facilities agricultural land
Ecological land	Woodland ecological landscapes	Forest land, Shrubbery, Other woodland, Orchard
	Pasture ecological landscapes	Population pasture, Other grassland, Natural pasture
	Other ecological landscapes	Saline-alkali land, Bare land, Inland tidal flats, Sandy land, Bare rock, Rivers, Reservoirs, Pits, Ditches

**Table 2 ijerph-17-01308-t002:** Evaluation indicators of the social–ecological production landscape resilience (SEPLR) in the Loess Plateau region.

Target Layer	Criterion Layer	Index Layer	Weight
Social–ecological production landscape resilience	Social system resilience (0.297)	Population density (***Z***_1_)	0.254
		Agricultural output value (***Z***_2_)	0.234
		Proportion of displaced population (***Z***_3_)	0.279
		Road density (***Z***_4_)	0.233
	Ecosystem resilience (0.343)	SHDI (***Z***_5_)	0.147
		Landscape connectivity index (***Z***_6_)	0.155
		NDVI (***Z***_7_)	0.194
		The application intensity of fertilizers (***Z***_8_)	0.177
		Precipitation erosivity (***Z***_9_)	0.156
		“Three types of land” areas (***Z***_10_)	0.171
	Production system resilience (0.360)	Slope (***Z***_11_)	0.119
		Elevation (***Z***_12_)	0.153
		Geological disaster risk (***Z***_13_)	0.157
		Grain yield (***Z***_14_)	0.132
		Cultivated land area (***Z***_15_)	0.159
		Amount of labor (***Z***_16_)	0.129
		Agricultural inputs (***Z***_17_)	0.151

SHDI: Shannon’s Diversity Index; NDVI: Normalized Difference Vegetation Index.

**Table 3 ijerph-17-01308-t003:** Criteria of SEPLs’ trade-off development zones.

Trade-Off Development Zones	Magic Cube Unit Combination
Synergistic promotion areas	(3,3,3) (3,3,2) (3,2,3) (2,3,3) (2,2,3) (2,3,2) (3,2,2) (2,2,2)
Ecological restoration areas	(1,3,3) (1,2,3) (1,3,2) (1,2,2)
Social development areas	(3,1,3) (3,1,2) (2,1,3) (2,1,2)
Production optimization areas	(3,3,1) (3,2,1) (2,3,1) (2,2,1)
Comprehensive remediation areas	(1,1,1) (1,1,2) (1,2,1) (2,1,1) (1,1,3) (1,3,1) (3,1,1)

**Table 4 ijerph-17-01308-t004:** Changes in the social–ecological production landscape pattern in Mizhi county from 2009 to 2018.

The Types of SEPLs	Area/km^2^		Change in Area/km^2^	Change Ratio/%
	2009	2018		
Agricultural production landscapes	511.77	625.79	114.02	18.22
Social living landscapes	38.03	39.44	1.41	3.58
Woodland ecological landscapes	275.28	140.83	−134.45	−95.47
Pasture ecological landscapes	336.53	338.64	2.11	0.62
Other ecological landscapes	17.24	34.15	16.91	49.52

**Table 5 ijerph-17-01308-t005:** Transfer matrix of SEPLs in Mizhi county from 2009 to 2018/km^2^.

2018	2009					
	Agricultural Production Landscapes	Social Living Landscapes	Woodland Ecological Landscapes	Pasture Ecological Landscapes	Other Ecological Landscapes	Total
**Agricultural production landscapes**	623.82	1.31	10.12	5.07	16.87	657.19
**Social living landscapes**	0	37.98	0	0	0	37.98
**Woodland ecological landscapes**	0.02	0.08	129.64	0.02	0.04	129.8
**Pasture ecological landscapes**	1.94	0.05	1.07	333.55	0.02	336.63
**Other ecological landscapes**	0	0.02	0	0	17.22	17.24
**Total**	625.78	39.44	140.83	338.64	34.15	1178.84

**Table 6 ijerph-17-01308-t006:** Top 5 resilience obstacle factors in different trade-off development areas during 2009–2018.

Trade-Off Development Zones	Number of Watersheds	Area Percentage	Top 5 Obstacle Factors				
			First Obstacle Factor	Second Obstacle Factor	Third Obstacle Factor	Fourth Obstacle Factor	Fifth Obstacle Factor
Synergistic promotion areas	94	27.26%	Amount of labor (***Z***_16_)	Application intensity of fertilizers (***Z***_8_)	Slope (***Z***_11_)	Agricultural output value (***Z***_2_)	Population density (***Z***_1_)
Ecological restoration areas	26	12.34%	Application intensity of fertilizers (***Z***_8_)	“Three types of land” areas (***Z***_10_)	Precipitation erosivity (***Z***_9_)	Road density (***Z***_4_)	Geological disaster risk (***Z***_13_)
Social development areas	31	11.74%	Agricultural output value (***Z***_2_)	Proportion of displaced population (***Z***_3_)	NDVI (***Z***_7_)	Amount of labor (***Z***_16_)	Grain yield (***Z***_14_)
Production optimization areas	54	28.58%	Cultivated land area (***Z***_15_)	Amount of labor (***Z***_16_)	Slope (***Z***_11_)	Road density (***Z***_4_)	Precipitation erosivity (***Z***_9_)
Comprehensive remediation areas	48	20.05%	NDVI (***Z***_7_)	Road density (***Z***_4_)	Agricultural output value (***Z***_2_)	Amount of labor (***Z***_16_)	“Three types of land” areas (***Z***_10_)

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
