# Peer review of "Evolutionary Characteristics and Trade-Offs’ Development of Social–Ecological Production Landscapes in the Loess Plateau Region from a Resilience Point of View: A Case Study in Mizhi County, China"

_ijerph, 2020, doi:10.3390/ijerph17041308_

Round 1

Reviewer 1 Report

 Line 66, here need a insertion of year-“Ciftciglu et al.”  The left map in Figure 1 needs a scale.  The figure 2 could be changed into colored one. The manuscript did give a good presentation of research results, but I would like suggest authors increase the space of the discussions.  Those literatures in Chinese needs the words of “in Chinese” added in the reference list.  It needs a more detailed description about how to calculate the index weight in Table 2.

Reviewer 2 Report

The manuscript is well-written and of reasonable interest. I have only a few comments and only one is major.

Table 6: This table needs to be reconfigured to fix column headers and rather than use the obstacle variable nomenclature, use the name of the variable as in the text. For example Z16 is amount of labor. This will make the table useful. At present, the table is meaningless.
